# Cost-effectiveness and budget impact analysis of PPV23 vaccination for the Malaysian Hajj pilgrims

**Farhana Aminuddin**[ID]*, **Nur Amalina Zaimi, Mohd Shaiful Jefri Mohd Nor Sham Kunusagaran, Mohd Shahri Bahari, Nor Zam Azihan Mohd Hassan**

Institute for Health Systems Research, Ministry of Health Malaysia, Shah Alam, Malaysia

* farhana.a@moh.gov.my

**Data Availability Statement:** All data generated and/ or analysed during the current study are either included in this published article or can be found in the referenced literature.

## Abstract

The potential occurrence of disease outbreaks during the hajj season is of great concern due to extreme congestion in a confined space. This promotes the acquisition, spread and transmission of pathogenic microorganisms and pneumococcal disease are one of the most frequent infections among Hajj pilgrims. This study aimed to assess the cost-effectiveness and budget impact of introducing the PPV23 to Malaysian Hajj pilgrims. A decision tree framework with a 1-year cycle length was adapted to evaluate the cost-effectiveness of a PPV23 vaccination program with no vaccination. The cost information was retrieved from the Lembaga Tabung Haji Malaysia (LTH) database. Vaccine effectiveness was based on the locally published data and the disease incidence specifically related to *Streptococcus pneumoniae* was based on a literature search. Analyses were conducted from the perspective of the provider: Ministry of Health and LTH Malaysia. The incremental cost-effectiveness ratios (ICER), cases averted, and net cost savings were estimated. Findings from this study showed that PPV23 vaccination for Malaysian Hajj pilgrims was cost-effective. The PPV23 vaccination programme has an ICER of MYR -449.3 (US$-110.95) per case averted. Based on the national threshold value of US$6,200-US$8,900 per capita, the base-case result shows that introduction of the PPV23 vaccine for Malaysian Hajj pilgrims is very cost-effective. Sensitivity analysis revealed parameters related to annual incidence and hospitalised cost of septicemia and disease without vaccination as the key drivers of the model outputs. Compared with no vaccination, the inclusion of PPV23 vaccination for Malaysian Hajj pilgrims was projected to result in a net cost saving of MYR59.6 million and 109,996 cases averted over 5 years period. The PPV23 vaccination program could substantially offer additional benefits in reducing the pneumococcal disease burden and healthcare cost. This could be of help for policymakers to consider the implementation of PPV23 vaccination for Malaysian performing hajj.

**Funding:** This study was supported/ funded by the Ministry of Health Malaysia (NMRR-17-471-34944). The funder had no role in study design, data collection and analysis, or preparation of the manuscript.

**Competing interests:** The authors have declared that no competing interests exist.

# Introduction

Millions of Muslims from more than 180 countries converge in Saudi Arabia to perform Hajj each year [1]. The overcrowding and extreme congestion due to confined spaces present a significant challenge to the health of Hajj pilgrims as they are more susceptible to acquiring and spreading infectious diseases, likes respiratory tract infections. Among the respiratory illnesses, Community-acquired pneumonia (CAP) is the primary cause of critical illness and the leading cause of hospital admission among Hajj pilgrims [2–4]. A broad array of pathogens are corresponding to respiratory disorders, with *Streptococcus pneumoniae* being the most common isolated from patients [5]. This pathogen could also cause diseases with varied in severity from otitis media to CAP, through invasive pneumococcal diseases (IPD) such as meningitis and septicemia. Individuals with chronic diseases, the elderly and infants are at risk of severe infection [6, 7].

A substantial proportion of Hajj pilgrims who arrived in Saudi are older adults aged >50 and >65 years with more than 50% and 25%, respectively [8]. Many Hajj pilgrims also have underlying medical conditions such as chronic lung disease, chronic heart disease, diabetes or other chronic conditions [4, 9, 10]. Hajj pilgrims, especially those of extreme age or with pre-existing medical conditions are at risk of pneumococcal disease, thus, consideration should be given to vaccinating these groups against pneumococcal disease. The ideal vaccine profile would be one that has an impact on pathogen transmission through a reduction in the amount of bacterial acquisition. The likely candidate, 23-valent polysaccharides pneumococcal vaccine (PPV23) was found to be effective and reported to reduce the risk of CAP and IPD in older adults ≥65 years [11, 12]. Yet, the cost-effectiveness of PPV23 among Hajj pilgrims is unknown.

Studies on economic evaluations such as the cost-effectiveness of pneumococcal vaccines among adults have been conducted in many countries with diverse outcomes [13, 14]. The PPV23 was found to be cost-effective among the elderly aged ≥60 years in several studies. According to Wolff et al [15], the PPV23 was reported to significantly reduce the burden of pneumococcal disease and is likely to be cost-effective for 75 year-olds in a Swedish setting. Besides, PPV23 was reported as cost-effective over the PCV13 among the elderly (≥65) in the Netherlands [16]. A previous study has also analysed the cost-effectiveness of PPV23 vaccination in different age groups, which PPV23 has been found most cost-effective in older adults aged ≥60 years [17]. The results, however, are difficult to translate to the Hajj pilgrims given the diverse population and unique event.

Respiratory tract infection was reported during the hajj season, which accounted for 74% of all medical illnesses [18] and 39% of all patients being reported hospitalised due to pneumonia [19]. For this, respiratory tract infection continues to exert a health and economic burden on a health system of a country. As a country with a majority of the Muslim population, thousands of Malaysian Muslims attended hajj each year. A study conducted by Deris et al. [20] reported that 90% of Malaysian Hajj pilgrims had at least one respiratory symptom. Another study conducted by Balkhy et al. [21] suggested that one in three Hajj pilgrims was estimated to experience respiratory symptoms. While healthcare-related costs of hospitalisation, medication and emergencies in Saudi are all covered by the Malaysian government, the most cost-effective and efficacious vaccine to prevent pneumococcal disease should be recommended and implemented.

Many countries are evolved in developing or have developed recommendations for pneumococcal vaccination for Hajj pilgrims including the Gulf countries (Qatar, Kuwait, United Arab Emirates, Bahrain and Oman), the USA, certain European countries (the UK, France and Germany) and Malaysia [22]. Although there is no recommendation for pneumococcal

vaccination for Hajj pilgrims by the Saudi Ministry of Health, it is recommended for those with a high risk of pneumococcal disease. To date, the Saudi Arabia authorities have only mandated meningococcal meningitis vaccination for Hajj pilgrims arriving from all countries [23, 24]. Besides, attendees from all countries, particularly those at higher risk of severe influenza disease are recommended to be vaccinated against seasonal influenza while those arriving from the countries at risk of yellow fever should be vaccinated against the yellow fever vaccine. In addition, Hajj pilgrims from the African meningitis belt are given ciprofloxacin upon arrival [24]. These requirements should be met by Hajj pilgrims following the health regulations set by the Saudi Ministry of Health to avert future calamity.

The policy on pneumococcal vaccination remains optional in many countries including Malaysia. Decisions about implementation of pneumococcal vaccine intervention for Hajj pilgrims requires a careful assessment at the country level, by considering the pneumococcal disease incidence, hospitalisation rates due to pneumococcal disease in Saudi, vaccine effectiveness and vaccination cost. Hence, this study was carried out to assess the cost-effectiveness and budget impact of introducing the PPV23 vaccination strategy versus no vaccination from the perspective of the Malaysian government. The findings from this study will provide evidence for policymakers if PPV23 is to be considered as an additional immunization programme for Malaysian Hajj pilgrims.

## Materials and methods

### Model structure

A decision tree model was adapted from Aljunid et al. [25] and developed with modification to determine the cost-effectiveness of introducing a PPV23 vaccination strategy for Malaysian performing hajj. Costs, measured in MYR, and health effects, refer to cases averted were assigned to each health state in the model. The base case model was applied to the 2017 Hajj pilgrims cohort (N = 40,837) and analysis was based on a one-year time horizon. The health states built into the model are shown in Fig 1 and the model considered the following disease categories; meningitis, pneumonia, otitis media, septicemia, sinusitis and death. All causes of pneumococcal disease were at risk of death. The impact of vaccination on costs was broken down into costs related to vaccination implementation and costs averted due to reduced healthcare burden. All costs were valued in Malaysian Ringgit in 2017 with a 3% discounted rate was used, where applicable. The health effects were quantified as cases averted due to vaccination. The analysis was performed from the provider's perspective.

### Epidemiological and disease data

Model inputs for epidemiological data including the health status background and disease burden of 2017 Malaysian Hajj pilgrims were obtained from the Lembaga Tabung Haji (LTH) Malaysia database, LTH-THIS. However, unavailable epidemiological and incidence of pneumococcal disease studies in Malaysia necessitated the use of data from other countries. Therefore, the probability of each pneumococcal disease incidence was obtained from the published literature, which was confined to a population group of Hajj pilgrims or the elderly (Table 1).

### Vaccine effectiveness

The PPV23 vaccine effectiveness has been derived based on available local data that could be applied in our model. Vaccine effectiveness against the pneumococcal disease was taken from a prospective cohort study conducted in 2015 by the Institute for Medical Research (IMR) Malaysia of 1,000 individuals aged 50 years and older performing hajj [26]. The study has

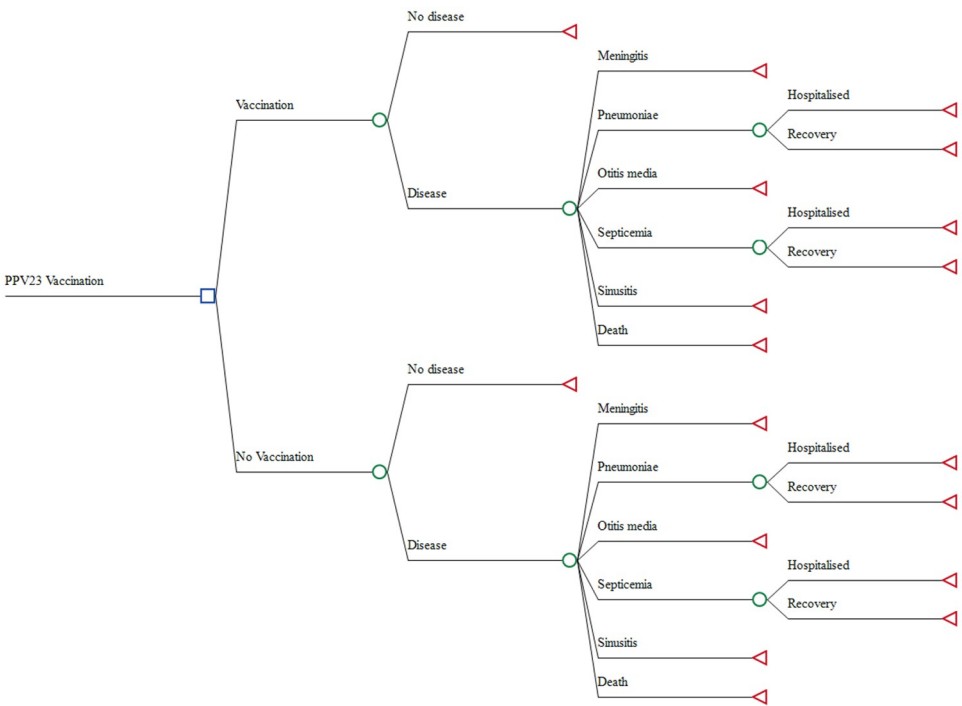

**Fig 1. A decision tree model, illustrating the two alternatives for PPV23 vaccination with the status quo of no vaccination.** The spectrum of pneumococcal infections included: meningitis, pneumonia, otitis media, septicemia, sinusitis and death.

reported that PPV23 vaccine effectiveness was 87.5%. For this, we used the vaccine effectiveness of ~88% and applied it in the base-case analysis.

## Costs estimation

This study considered different methods to estimate the outpatient and inpatient costs given the lack of data availability such as individual patients' medical visits and admission records.

**Table 1. Probabilities and estimates associated with pneumococcal disease.**

| Variables | Base-case value | Range for sensitivity analysisΨ | Distribution | Source |
|---|---|---|---|---|
| Incidence rates of pneumococcal disease: | | | | |
| With vaccination | 0.120 | 0.090–0.150 | Beta | Calculated from [26] |
| Without vaccination | 0.620 | 0.465–0.775 | Beta | Calculated from [26] |
| Incidence rates of pneumococcal outcome: | | | | |
| Meningitis | 0.007 | 0.005–0.009 | Beta | [27] |
| Pneumonia | 0.230 | 0.173–0.288 | Beta | [2] |
| Hospitalised pneumonia | 0.017 | 0.013–0.021 | Beta | Calculated from [25] |
| Otitis media | 0.200 | 0.150–0.250 | Beta | [28] |
| Septicemia | 0.250 | 0.188–0.313 | Beta | [29] |
| Hospitalised septicemia | 0.059 | 0.075–0.125 | Beta | Calculated from [25] |
| Sinusitis | 0.100 | 0.075–0.125 | Beta | [30] |
| Number of Malaysian Hajj pilgrimsΨΨ | 40,837 | 30,628–51,046 | Normal | LTH-THIS |

ΨA range of ±25% of the base-case value was used when reported data were not available.
ΨΨNumber of Malaysian Hajj pilgrims visited Saudi Arabia in 2017.

Outpatient and inpatient care costs included the cost of facility rentals, vehicle rentals, utilities, drugs, and staff allowances used during pilgrim's visits to the clinic and hospital admission. A top-down costing approach for outpatient was applied to calculate the operational costs incurred by the Ministry of Health (MOH) and LTH Malaysia for the management and treatment of pneumococcal disease based on actual expenditures in 2017 (Table 2). The costs for both clinics and hospitals were estimated using different apportion techniques; the total costs of the vehicles, electricity, water, salary, and petrol were equally distributed (50%:50%) between clinics and hospitals. While 60%:40% were allocated for drugs and equipment between clinics and hospitals, respectively. The apportionment of resources was by the experts' opinion based on the pattern of health care utilisation in clinics and hospitals. The cost of outpatient care was calculated based on the average cost of treatment per Hajj pilgrims per visit.

The cost of hospitalisation for Malaysian Hajj Pilgrims in Saudi Arabia was adapted from the Malaysian Diagnosis-related Group Case-mix Systems (My-DRG) costing data under a category of Severity 1 Level as a baseline to estimate the minimum cost of hospitalisation in Saudi Arabia (Table 3). This is due to limitations of data such as facilities floor space and the admission ratio of pneumococcal-related disease to other diseases. The costing data is then adjusted to the Purchasing Power Parity (PPP) per capita ratio between Saudi Arabia (US $47,309.132) and Malaysia (US$26,661.188) in 2017 [31].

Because there is no national vaccine cost data available, the PPV23 vaccine's unit price was adopted and derived from Mo et al [32]. All costs of vaccination were calculated in Malaysian Ringgit (MYR) for the year 2017 (USD 1 = RM4.30) after being adjusted to an average annual inflation rate of 3%. For this, the cost of vaccination used for the analysis was estimated at RM130.80 per dose. The costs of pneumococcal disease treatment are tabulated in Table 4.

**Table 2. Costs estimate per year for outpatient care to treat pneumococcal disease based on actual expenditures in 2017.**

|  | Outpatient care (MYR) |
| --- | --- |
| Rentals |  |
| Clinics | 3,074,400 |
| Vehicles | 216,978 |
| Utility bills |  |
| Electricity | 54,000 |
| Water | 39,600 |
| Drugs & consumables |  |
| Purchasing cost in Malaysia | 1,632,000 |
| Delivery cost to Saudi Arabia | 138,000 |
| Purchasing cost in Saudi Arabia | 90,000 |
| Salary & allowances | 1,128,731.50 |
| Others |  |
| Petrol for vehicles | 5,000 |
| Patient's foods | 0 |
| **Total costs** | **6,378,709.50** |
| Total no. of visits | 54,495 |
| Total costs per visits | 117.05 |
| Frequency of visits per person$^{\Psi}$ | 3 |
| **Outpatient** |  |
| Cost per person | **351.15** |

$^{\Psi}$An average of 3 visits per one treatment per person was assumed, which was based on LTH expert opinion.

**Table 3. Cost estimations for inpatient care to treat pneumococcal disease in 2017 based on My-DRG.**

| Inpatient care (per admission) | Malaysia (DRG-Severity 1) (MYR) | Saudi (MYR)$ |
|---|---|---|
| Pneumonia | 5,435.88 | 9,944.92 |
| Septicemia | 9,338.95 | 17,085.57 |
| Meningitis | 11,920.75 | 21,808.96 |

$Estimated based on PPP per capita ratio between Malaysia and Saudi in 2017 [31] and a 3% annual inflation rate was applied.

## Cost-effectiveness analysis

Incremental cost-effectiveness ratios (ICERs), which are a measure of cost-effectiveness, was reported as cost per case averted. The ICERs were determined by dividing the treatment costs of pneumococcal disease (with or without PPV23 vaccination) by the incremental effectiveness (measured as cases averted) between the two interventions. ICER was reported as incremental cost per case averted. The cost savings due to vaccine prevented cases were also reported. A national cost-effectiveness threshold was used, whereby an ICER between 0.90 and 1.28 (MYR19,929 and MYR28,470 or US$6,200-US$8,900 per capita) is considered cost-effective and very cost-effective when the ICER value is below the threshold range [33].

## Uncertainty analysis

**One-way sensitivity analysis.** Sensitivity analyses were performed to assess the influence of input parameters on the cost-effectiveness results. A one-way sensitivity analysis (vaccination vs no vaccination) was conducted to test the robustness of the model and to find the most significant variables influencing the cost-effectiveness of the PPV23 vaccination programme. The result of the one-way sensitivity analysis is presented in a Tornado diagram. The values for each base-case model parameter were obtained from the relevant literature and a range of ±25% of the base-case values was used when reported data were not available (see Tables 1 and 4).

**Probabilistic sensitivity analysis.** Bayesian methods such as probabilistic sensitivity analysis (PSA) are often performed to evaluate the impact of the uncertainty in all of the variables simultaneously [34, 35]. PSA was done by assigning the model parameters with appropriate distributions model. In this study, the model was simulated 1,000 times by allowing the probabilities and costs parameters to vary for uncertainties assessment. All included probabilities were subject to a beta distribution and the costs parameters followed a gamma distribution.

**Table 4. The costs of vaccine per dose and pneumococcal disease.**

| Variables | Base-case estimate (MYR) | Range for sensitivity analysisΨ | Distribution | Source |
|---|---|---|---|---|
| Cost of vaccination | 130.80 | 98.10–163.50 | Gamma | [32] |
| Cost of treatment | | | | |
| Meningitis | 21,808.96 | 16,356.72–27,261.20 | Gamma | My-DRG |
| Pneumonia | 9,944.92 | 7,458.69–12,431.15 | Gamma | My-DRG |
| Otitis media | 351.15 | 263.36–438.94 | Gamma | Calculated$ |
| Septicemia | 17,085.57 | 12,814.18–21,356.96 | Gamma | My-DRG |
| Sinusitis | 351.15 | 263.36–438.94 | Gamma | Calculated$ |

$Using the cost data obtained from the LTH-THIS database.

The cost-effectiveness plane scatterplot and Cost-Effectiveness Acceptability Curve (CEAC) were visualised to test the stability of the model results.

### Budget impact analysis

BIA was calculated to project a financial burden for future health expenditures if the PPV23 vaccination programme is introduced to Malaysian Hajj pilgrims compared to no vaccination. The total budget of the vaccination programme was calculated by multiplying the number of Hajj pilgrims per year by the cost of the vaccine per dose. We assumed 100% of Hajj pilgrims will be vaccinated each year, and the cost of vaccination is adjusted to the annual inflation rate of 3%. The treatment costs averted due to the PPV23 vaccination programme was presented to obtain the net incremental cost of the programme.

### Ethics statement

Ethics approval for the study was granted by the Medical Research and Ethics Committee (MREC), Ministry of Health Malaysia with reference number NMRR-17-471-34944. Informed consent was not obtained as this study involved secondary data analysis using available data, thus it was deemed not necessary by the Ethics Committee.

## Results

### Base case analysis

The results from the cost-effectiveness analyses of PPV23 vaccination are presented in Table 5. It was estimated that vaccination with a PPV23 would prevent 35,937 pneumococcal-related cases compared with no vaccination strategy for the Malaysian Hajj pilgrims of 40,837 in 2017. The PPV23 vaccination would incur a cost of MYR8.8 million while the Malaysian health system would incur a total cost of MYR18 million for the study cohort due to pneumococcal disease without vaccination in place. In terms of cost-effectiveness, the PPV23 vaccination programme has an ICER of MYR -449.3 per case averted compared with a status quo of no vaccination. Based on the national threshold value of US$6,200-US$8,900 per capita, the base-case result shows that introduction of the PPV23 vaccine for Malaysian Hajj pilgrims is very cost-effective.

### Uncertainty analyses

**One-way sensitivity analysis.** The results of one-way sensitivity analysis are shown in the tornado diagram (Fig 2), which graphically represent how variations in each parameter affect

**Table 5. Cost-effectiveness analysis of the PPV23 vaccination programme in a cohort of Malaysian Hajj pilgrims.**

| Variable | No vaccination | Vaccination | Incremental events or costs (averted) | Incremental events or costs (incurred) |
|---|---|---|---|---|
| Cost of vaccination | - | 5,341,479.60 | - | 5,341,479.60 |
| Cost of Meningitis | 3,865,368.21 | 748,114.50 | 3,117,253.71 | - |
| Cost of Pneumonia | 2,994,626.57 | 579,605.14 | 2,415,021.43 | - |
| Cost of Otitis media | 1,778,359.21 | 344,157.90 | 1,434,201.31 | - |
| Cost of Septicemia | 8,472,228.71 | 1,639,786.20 | 6,832,442.51 | - |
| Cost of Sinusitis | 888,754.30 | 172,078.96 | 716,675.34 | - |
| Total Costs | 17,999,337.00 | 8,825,222.30 | 9,174,114.70 | - |
| Effects (case averted) | 15,518 | 35,937 | - | 20,419 |
| ICER (per case averted) | - | -449.30 | - | - |

All costs were measured in Malaysian Ringgit (MYR).

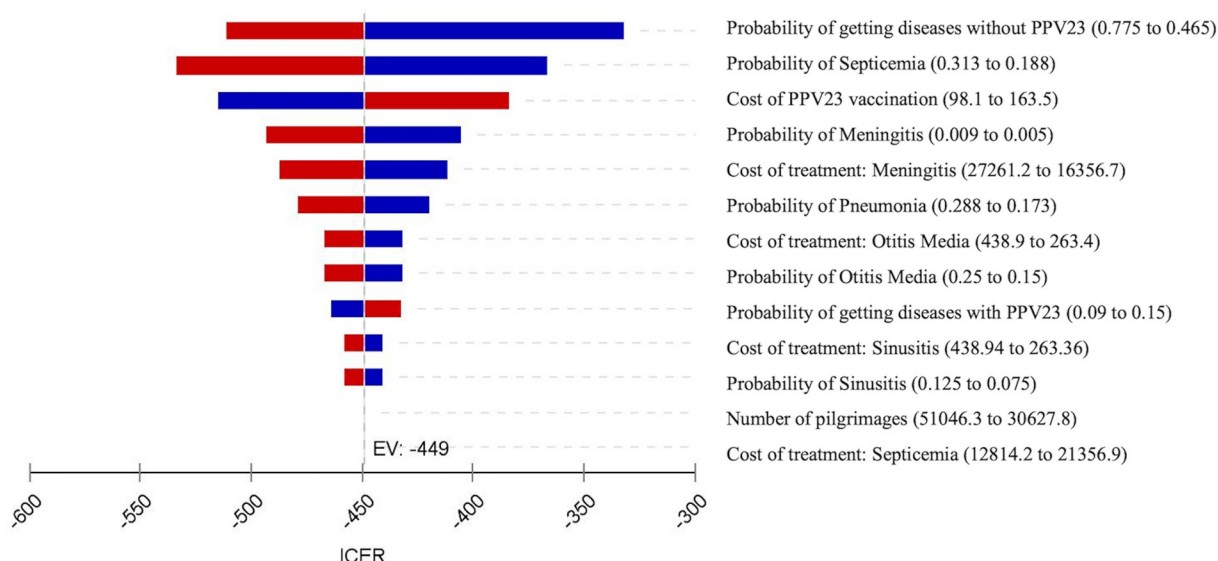

**Fig 2. Tornado chart presenting a one-way sensitivity analysis for PPV23 vaccination.** The tails of each bar represent the differences in ICERs with low (left) and high (right) parameter values. The maximum and minimum values for each variable are presented in brackets. EV: Expected value.

the outcome (ICERs). The ICERs of PPV23 vaccination versus a no vaccination strategy ranged from MYR-533.63 to -331.77. The annual incidence of getting the disease without vaccination and septicemia are the most influential parameters in the model output. Another parameter with a remarkable impact on results was the PPV23 vaccination cost. These variables were also identified as the key drivers of the model outputs. The parameters, such as the treatment cost of meningitis, otitis media and sinusitis, only have medium and small impacts. The results demonstrated that this model was also sensitive to the cost of the vaccine and the probabilities of getting the disease with vaccination, which showed a positive relationship.

**Probabilistic sensitivity analysis.** A Monte Carlo simulation with 1,000 iterations was done by comparing the PPV23 vaccination programme with no vaccination (Fig 3). This

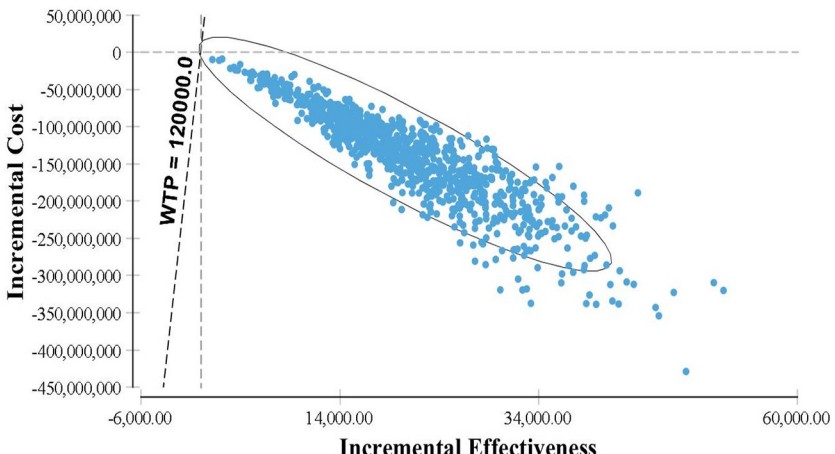

**Fig 3. Scatter plot showing results of Monte Carlo probabilistic sensitivity analysis for PPV23 vaccination programme vs no vaccination.**

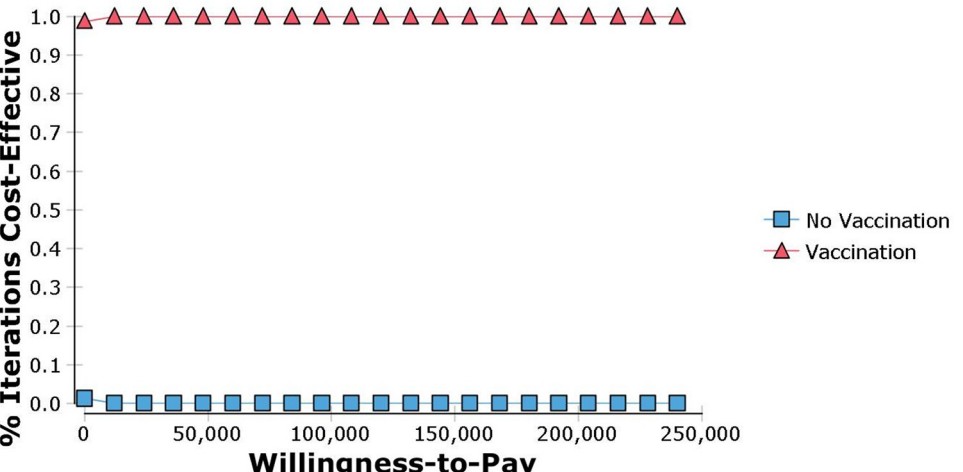

**Fig 4. Cost-effectiveness acceptability curve showing the probability that vaccination is cost-effective compared to no vaccination over a range of values.**

shows that the cost-effectiveness probabilities represented as a cloud of points lie in the South Eastern (SE) quadrant plane (97%) after considering all the uncertainties in the analysis. This corresponds that the introduction of the PPV23 vaccine provides additional health benefits (incremental effectiveness increases) at a lower cost (incremental cost decreases) despite the variation of parameters. Fig 4 illustrates the cost-effectiveness acceptability curve to assess the threshold of willingness-to-pay (WTP) for the PPV23 vaccine in Malaysia. It demonstrates that vaccination was more effective compared to no vaccination throughout various WTP threshold values up to MYR 240,000.

## Budget impact analysis

The budget impact analysis of the PPV23 vaccine was projected as tabulated in Table 6. The total expenditure on vaccinating a cohort of Hajj pilgrims for PPV23 vaccination during the year 2021–2025 was MYR34.7 million. While in the current scenario, without PPV23 vaccination, Malaysia's healthcare spending on pneumococcal disease among Hajj pilgrims for a 5-year (2021–2025) total is estimated to be MYR117.1 million. In this base case analysis, the number of cases averted grows over time due to the progressive increase in the number of Hajj pilgrims annually. Inclusion of PPV23 vaccination for a cohort of Hajj pilgrims will result in a net cost saving of MYR59.6 million and 110,996 cases averted over 5 years (2021–2025) period after discounting. From these findings, the introduction of the PPV23 vaccine for Malaysian Hajj pilgrims will affect the healthcare budget by almost 50% cost reduction yearly.

## Discussion

We created a decision tree framework to evaluate the cost-effectiveness of the pneumococcal vaccine, PPV23 for Malaysian Hajj pilgrims compared to the current status of no vaccination. The results presented here demonstrated that the implementation of a PPV23 vaccination for Malaysian Hajj pilgrims can effectively reduce health (cases averted) and economic burden (cost savings) associated with pneumococcal diseases. Under the base case scenario, PPV23 could be considered highly cost-effective, with an ICER of MYR -449.3 per case averted. This is in line with other modelling studies conducted in several countries like the Netherlands,

**Table 6. Budget impact analysis of PPV23 vaccination.**

| | 2017 | 2018 | 2019 | 2020 | 2021 | 2022 | 2023 | 2024 | 2025 | Total (5 years results)ΨΨ |
|---|---|---|---|---|---|---|---|---|---|---|
| Number of Hajj pilgrimsΨ | 40,837 | 41,409 | 41,988 | 42,576 | 43,172 | 43,777 | 44,390 | 45,011 | 45,641 | |
| No vaccination | | | | | | | | | | |
| Number of cases | 25,319 | 25,673 | 26,033 | 26,397 | 26,767 | 27,142 | 27,522 | 27,907 | 28,298 | |
| Cost of treatment (MYR in million) | 18.0 | 18.8 | 19.6 | 20.5 | 21.4 | 22.4 | 23.4 | 24.4 | 25.5 | 117.1 |
| Vaccination | | | | | | | | | | |
| Number of pneumococcal-related cases | 4,900 | 4,969 | 5,039 | 5,109 | 5,181 | 5,253 | 5,327 | 5,401 | 5,477 | |
| Cost of vaccination/ dose (MYR) | 130.8 | 134.7 | 138.8 | 142.9 | 147.2 | 151.6 | 156.2 | 160.9 | 165.7 | |
| Cost of vaccination (MYR in million) | 5.3 | 5.6 | 5.8 | 6.1 | 6.4 | 6.6 | 6.9 | 7.2 | 7.6 | 34.7 |
| Cost of treatment (MYR in million) | 3.5 | 3.6 | 3.8 | 4.0 | 4.1 | 4.3 | 4.5 | 4.7 | 4.9 | 22.5 |
| Total cost (vaccination + treatment) (MYR in million) | 8.8 | 9.2 | 9.6 | 10.1 | 10.5 | 10.9 | 11.4 | 11.9 | 12.5 | 57.2 |
| Number of cases averted | 20,419 | 20,704 | 20,994 | 21,288 | 21,586 | 21,888 | 22,195 | 22,506 | 22,821 | 110,996 |
| Net cost-saving (MYR in million) | 9.2 | 9.6 | 10.0 | 10.4 | 10.9 | 11.4 | 11.9 | 12.4 | 13.0 | 59.6 |

The cost of vaccination was adopted from Mo et al [32] and adjusted to annual inflation rate of 3%.

ΨNumber of Hajj pilgrims was calculated based on Malaysian population growth of 1.4% [36].

ΨΨ5 years total costs was calculated from year 2021 to 2025.

Belgium, Turkey, Colombia and Brazil also found PPV23 to be a cost-effective intervention [17, 37–40]. An additional 110,966 cases of pneumococcal disease could be averted with an estimated MYR 59.6 million cost-saving. This is in line with other studies, which reported the cost-saving intervention [17, 40–43]. However, further comparison with the international literature is not possible due to different assumptions on model structures, as well as parameters included such as time horizon, vaccine price, vaccine effectiveness and incidence of related pneumococcal disease.

The hypothetical analysis conducted in this study shows that the implementation of the PPV23 vaccine to a 2017 cohort of Malaysian Hajj pilgrims has the potential to reduce the pneumococcal disease burden by 56.8%. The costs to vaccinate a cohort of Malaysian Hajj pilgrims (MYR8.8 million) are much smaller than the medical costs of pneumococcal disease (MYR18 million). In this base case analysis, the estimated cost associated with pneumococcal disease was reduced by 46.8%. Due to the unavailable cost of PPV23, the best available cost of vaccine in the region was used in the analysis [32]. This may potentially produce variability with the data included in the model. Hence, to ascertain a programme is still a cost-effective option, uncertainty analysis (one-way and probabilistic sensitivity) was conducted. This implies that a relevant variation of the input parameters would not alter the conclusions of this cost-effectiveness analysis study. The important uncertainties are presented through one-way sensitivity analysis, which showed that the probability of getting the pneumococcal disease without vaccination affects ICER the most followed by the probability of septicemia. Most notably, the PPV23 vaccination cost is also a major player driving the ICERs. Certainly, the current cost of vaccines could potentially avoid large numbers of pneumococcal diseases and be regarded as a cost-saving measure. Besides, probabilistic sensitivity analyses showed that ICERs were predominantly cost-effective, with a very small percentage of ICERs being dominated.

The introduction of the PPV23 vaccine has increased interest in preventing pneumococcal disease. In most developed countries, the PPV23 is recommended for elderly adults [44–46]. This could be due to its broad serotype coverage, which has been estimated to be 84% among adults [47]. In relation to hajj, recommendation on the uptake of the PPV23 vaccine prior to

performing the hajj has been evolved in certain countries, including Malaysia. However, there is a lack of consensus as to whether the recommendation of the PPV23 is for all adults or specifically for the Hajj pilgrims population. In this analysis, we focused on the PPV23 intervention among Hajj pilgrims, which is normally dominated by the elderly population. According to a study conducted by Eilers et al. [48], pneumococcal disease is considered the most severe and most relevant disease to vaccinate among the elderly. Perhaps, assumption of a 100% uptake of PPV23 among the Hajj pilgrims in this study could be considered as a potential strategy to vaccinate Malaysians who wish to perform hajj in the future. Although this study did not consider payment options from the Hajj pilgrims or co-payment between the government and Hajj pilgrims, full subsidisation from the Malaysian government could still offer a cost-saving measure and ultimately reduce disease and economic burden from a healthcare system perspective.

To the authors' knowledge, this is the first to look into the cost-effectiveness of implementing pneumococcal vaccination using the PPV23 vaccine, among Malaysian Hajj pilgrims, in terms of both health outcomes and budget impact. Our study has certain strengths. The PPV23 effectiveness was based on the prospective cohort that involved vaccinated and a non-vaccinated group of Malaysian Hajj pilgrims in 2015 [24], which clearly showed that PPV23 is efficacious in the prevention of pneumococcal pneumonia. Besides, findings from the budget impact analysis enable the policymakers to know the total budget, incremental costs, and costs saving by implementing the PPV23 vaccination. This will allow improving the allocation of scarce healthcare resources and would facilitate the practice of evidence-based reimbursement decision making. Our findings also add to the limited evidence available regarding the cost-effectiveness of pneumococcal vaccination among Hajj pilgrims. This could be of help to contribute to the knowledge generation of future pneumococcal vaccination programmes among the pool of Hajj pilgrims. We conducted this study from the provider perspective, which the medical costs incurred by the Malaysian Government. This study perhaps could provide useful information to policymakers for the planning and development of a PPV23 vaccination programme for Malaysian Hajj pilgrims. It could also be used to allocate scarce resources for the health of Hajj pilgrims, which includes qualified personnel and operational costs of hospitals and clinics in Saudi.

Notwithstanding a thorough analysis, this study still experienced some limitations. First, data used in this study were subjected to availability and the analyses were restricted based on the best available data shared. The data inputs into the model were obtained from the literature (non-Malaysian population), which is confined to Hajj pilgrims or the elderly. This is due to the absence of published local data and potentially makes estimates in the model uncertain. However, the one-way and probabilistic sensitivity analyses were conducted to overcome this constraint using a plausible range of uncertain parameters. Second, the time frame or duration of this study was set at one year, within the year 2017 with no further follow up of the Hajj pilgrims, as the pilgrimage typically occurs only once a year. Third, only the operational costs at Saudi Arabia health facilities were considered and included in this cost-effectiveness analysis, thus, post-hajj operational costs are excluded, given that Hajj pilgrims may develop symptoms and seek treatment after they returned to Malaysia. Besides, the cost-effectiveness results were estimated using case averted, which further limit the comparison to the WTP threshold and with other studies incorporated an estimation using DALYs or QALYs. Lastly, the context of this study was confined to a one-off exposure to streptococcal infection within the specified time frame. For this, herd immunity was not included at the data analysis stage because the evidence on this topic was not available, particularly in a study in which vaccination was only given to a relatively small population such as the Hajj pilgrims. Despite these shortcomings, our analysis provides enlightenment into the potential of PPV23 to reduce the burden of pneumococcal disease.

## Conclusions

This study assesses the cost-effectiveness and budget impact of a PPV23 vaccination programme for Malaysian Hajj pilgrims, compared with no vaccination. The results presented here demonstrated that PPV23 would be a cost-effective measure to reduce the economic and health burden of pneumococcal disease. Besides, it is suggested that this vaccination programme would avoid a significant number of pneumococcal cases over 5 years and would be a cost-saving measure from a health system perspective. At the current rate, it is evident that the health and economic needs of the Hajj pilgrims pool will be enormous. In particular, costs related to pneumococcal disease treatment for this specific group will be more demanding in the future. Certainly, this necessitates a cost-effective and efficient approach. Thus, on the basis of the study findings, it could be suggested that PPV23 should be implemented as the vaccination programme for Malaysian performing the hajj, especially those at risk of pneumococcal disease.

## Acknowledgments

The authors wish to thank the Director-General of Health, Malaysia that made this publication possible. We also acknowledge Lembaga Tabung Haji Malaysia and the Institute for Medical Research, Ministry of Health Malaysia for providing data for this study. We would like to thank the reviewers for their thoughtful comments and suggestions.

## Author Contributions

**Conceptualization:** Farhana Aminuddin, Nor Zam Azihan Mohd Hassan.

**Data curation:** Farhana Aminuddin.

**Formal analysis:** Farhana Aminuddin, Nor Zam Azihan Mohd Hassan.

**Investigation:** Nur Amalina Zaimi, Mohd Shahri Bahari.

**Methodology:** Farhana Aminuddin, Nur Amalina Zaimi, Mohd Shaiful Jefri Mohd Nor Sham Kunusagaran.

**Project administration:** Nur Amalina Zaimi.

**Supervision:** Nor Zam Azihan Mohd Hassan.

**Validation:** Mohd Shaiful Jefri Mohd Nor Sham Kunusagaran, Mohd Shahri Bahari.

**Visualization:** Nor Zam Azihan Mohd Hassan.

**Writing – original draft:** Farhana Aminuddin.

**Writing – review & editing:** Nur Amalina Zaimi, Mohd Shaiful Jefri Mohd Nor Sham Kunusagaran, Mohd Shahri Bahari, Nor Zam Azihan Mohd Hassan.

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
