## [Decision Letter · Decision Letter 0]

8 Dec 2021

PONE-D-21-27712Cost-effectiveness and budget impact analysis of PPV23 vaccination for the Malaysian Hajj pilgrimsPLOS ONE

Dear Dr. Aminuddin,

Thank you for submitting your manuscript to PLOS ONE. After careful consideration, we feel that it has merit but does not fully meet PLOS ONE’s publication criteria as it currently stands. Therefore, we invite you to submit a revised version of the manuscript that addresses the points raised during the review process.

We look forward to receiving your revised manuscript.

Kind regards,

M. Mahmud Khan

Academic Editor

PLOS ONE

Journal Requirements:

Reviewers' comments:

Reviewer's Responses to Questions

**Comments to the Author**

1. Is the manuscript technically sound, and do the data support the conclusions?

Reviewer #1: Yes

Reviewer #2: Yes

2. Has the statistical analysis been performed appropriately and rigorously? 

Reviewer #1: Yes

Reviewer #2: Yes

3. Have the authors made all data underlying the findings in their manuscript fully available?

Reviewer #1: Yes

Reviewer #2: Yes

4. Is the manuscript presented in an intelligible fashion and written in standard English?

Reviewer #1: Yes

Reviewer #2: Yes

5. Review Comments to the Author

Reviewer #1: This work results will be helpful not just to the Malaysian Hajj visitors , but to all Hajj visitors from other counties. The only thing that I need you to review please is the vaccination required by the Saudi government which change every year , and different from country to a country based on the public health status of the visitor country ,

Reviewer #2: he manuscript has been prepared in accordance with the results of the study and can be concluded well. For the writing of the words Hajj and Pilgrims should be consistent, some are written Hajj Pilgrims, some are written Hajj only or Pilgrims only.

For the figure, the decision tree model doesn't appear, I don't know if there's a difference between the script I received and the original. In my opinion, the description on probabilistic sensitivity analysis is incomplete and unclear, so it needs to be described properly and can be added with reference materials related to this topic.

6. PLOS authors have the option to publish the peer review history of their article (what does this mean?). If published, this will include your full peer review and any attached files.

Reviewer #1: No

Reviewer #2: **Yes: **Setya Haksama

---

## [Author Response · Author response to Decision Letter 0]

4 Jan 2022

Response to Reviewers

Dear Professor Mahmud Khan,

Thank you for giving us the opportunity to submit a revised draft of the manuscript “Cost-effectiveness and budget impact analysis of PPV23 vaccination for the Malaysian Hajj pilgrims”. We appreciate the time and effort that you and the reviewers dedicated to providing feedback on our manuscript and are grateful for the insightful comments. We have revised the manuscript accordingly (incorporate all suggestions) and provided specific answers below, written in red font. All changes made in the main document are marked using track changes. 

Academic editor:

1. Please ensure that your manuscript meets PLOS ONE’s style requirements, including those for file naming. 

We have thoroughly checked the manuscript and prepared it following PLOS ONE’s style requirements.

We have added a full ethics statement in the Methods section line 251. In this statement, the ethics committee is mentioned together with informed consent which is not necessary for this study. 

3. Please review your reference list to ensure that it is complete and correct. If you have cited papers that have been retracted, please include the rationale for doing so in the manuscript text, or remove these references and replace them with relevant current references. 

All included references were checked for completeness and correctness. There are no retracted papers cited in this study.

Reviewers:

Reviewer#1 

Comment: This work results will be helpful not just to the Malaysian Hajj Visitors, but to all Hajj visitors from other countries. The only thing that I need you to review please is the vaccination required by the Saudi government which change every year, and different from country to a country based on the public health status of the visitor country.

Answer: We have reviewed and added several health requirements that must be met by Hajj pilgrims from different countries, where the health regulations are set by the Saudi Ministry of Health (Refer to introduction section, lines 108-117). This information can be obtained from the Embassy of the Kingdom of Saudi Arabia. 

Reviewer#2

Comment: The manuscript has been prepared in accordance with the results of the study and can be concluded well. For the writing of the words, Hajj and Pilgrims should be consistent, some are written Hajj Pilgrims, some are written Hajj only or Pilgrims only.

Answer: We go through the manuscript and the word ‘pilgrims’ won’t appear alone instead, written as Hajj pilgrims. And wherever relevant, the word hajj could be appeared alone according to the context of the sentences. (Hajj pilgrims are referred to as the Muslim population while hajj is the rituals performed by these groups).

Comment: For the figure, the decision tree model doesn’t appear, I don’t know if there’s a difference between the script I received and the original. 

Answer: We apologise for the inconvenience caused. However, I guarantee the decision tree figure that you received is similar to the original one. The tree model was created based on a prior study conducted by Aljunid et al., 2014. A slight modification of the model was made based on data availability which was confined to the Hajj pilgrims population. 

Comment: In my opinion, the description on probabilistic sensitivity analysis is incomplete and unclear, so it needs to be described properly and can be added with reference materials related to this topic.

Answer: Changes were made to the description of probabilistic sensitivity analysis (refer lines 234-241) to make it clear and understandable.

---

## [Editor Report · Decision Letter 1]

10 Jan 2022

Cost-Effectiveness and Budget Impact Analysis of PPV23 Vaccination for the Malaysian Hajj Pilgrims

PONE-D-21-27712R1

Dear Dr. Aminuddin,

We’re pleased to inform you that your manuscript has been judged scientifically suitable for publication and will be formally accepted for publication once it meets all outstanding technical requirements.

Kind regards,

M. Mahmud Khan

Academic Editor

PLOS ONE
---

## [Editor Report · Acceptance letter]

12 Jan 2022

PONE-D-21-27712R1 

Cost-Effectiveness and Budget Impact Analysis of PPV23 Vaccination for the Malaysian Hajj Pilgrims 

Dear Dr. Aminuddin:

I'm pleased to inform you that your manuscript has been deemed suitable for publication in PLOS ONE. Congratulations! Your manuscript is now with our production department. 

Kind regards, 

on behalf of

Dr. M. Mahmud Khan 

Academic Editor

PLOS ONE